# The Maximum Flywheel Load: A Novel Index to Monitor Loading Intensity of Flywheel Devices

**DOI:** 10.3390/s21238124

**Published:** 2021-12-04

**Authors:** Alejandro Muñoz-López, Pablo Floría, Borja Sañudo, Javier Pecci, Jorge Carmona Pérez, Marco Pozzo

**Affiliations:** 1Departamento de Motricidad Humana y Rendimiento Deportivo, University of Seville, 41013 Seville, Spain; jorgecarmonap5@gmail.com; 2Physical Performance and Sports Research Center, Universidad Pablo de Olavide, 41013 Seville, Spain; pfloriam@upo.es; 3Department of Physical Education, University of Seville, 41013 Seville, Spain; bsancor@us.es (B.S.); javipecci99@gmail.com (J.P.); 4SmartCoach Europe AB, 118 20 Stockholm, Sweden; marco.pozzo@smartcoach.tech

**Keywords:** programming, strength, training, eccentric overload, force, speed, force-velocity profile

## Abstract

Background: The main aim of this study was (1) to find an index to monitor the loading intensity of flywheel resistance training, and (2) to study the differences in the relative intensity workload spectrum between the FW-load and ISO-load. Methods: twenty-one males participated in the study. Subjects executed an incremental loading test in the squat exercise using a Smith machine (ISO-load) or a flywheel device (FW-load). We studied different association models between speed, power, acceleration, and force, and each moment of inertia was used to find an index for FW-load. In addition, we tested the differences between relative workloads among load conditions using a two-way repeated-measures test. Results: the highest r2 was observed using a logarithmic fitting model between the mean angular acceleration and moment of inertia. The intersection with the x-axis resulted in an index (maximum flywheel load, MFL) that represents a theoretical individual maximal load that can be used. The ISO-load showed greater speed, acceleration, and power outcomes at any relative workload (%MFL vs. % maximum repetition). However, from 45% of the relative workload, FW-load showed higher vertical forces. Conclusions: MFL can be easily computed using a logarithmic model between the mean angular acceleration and moment of inertia to characterize the maximum theoretical loading intensity in the flywheel squat.

## 1. Introduction

In strength training (RT), load indicates the amount of external resistance during exercise execution [1]. In weight training (ISO-load), exercise intensity can be measured efficiently by execution speed [2]. However, traditionally, load magnitude (i.e., weight lifted) has been widely used. In ISO-load, the weight that can be lifted only once is known as the one maximum repetition (1RM) weight. From the 1RM, different training zones can be defined, based on intervals of relative load (i.e., %1RM) [3]. Consequently, %1RM can be used to individualize an RT program in order to optimize its training effects [1,2]. 

Over the last decade, there has been increasing interest in the use of flywheel resistance training devices (FRTD) in RT programs [4]. An FRTD is a piece of exercise equipment where the trainee pulls a rope (or strap) wound onto the shaft of a spinning flywheel. This device allows maximal voluntary force exertion during the whole concentric phase of the movement, converting the trainee’s work into rotational kinetic energy stored in the flywheel, which is returned during the subsequent eccentric phase [5]. A recent review comparing load in FRTDs (FW-load) with ISO-load showed higher levels of hypertrophy in subjects using FW-load [6]. However, there are still some inconsistencies when the two loading conditions are compared [7]. One problem when making comparisons is the absence of a maximal load index in FW-load, equivalent to that of 1RM in ISO-load; this is due to the physical working principle of FRTDs, where the load is fully inertial, so that no matter how large the inertia and how small the exerted force, the flywheel can always be spun.

The comparison of differences in training effects between FW-load and ISO-load has been of great interest [8,9,10,11,12,13,14]. In these studies, a higher mechanical demand on the knee extensors [9], increased metabolic demands [10], comparable muscle hypertrophy [13], and higher eccentric muscle activity [8] were observed when FW-loads were used in comparison with ISO-loads. However, many past studies lacked a sound criterion to match loading in the two training devices [9,10,13]. To match the external load to compare FW-load against ISO-load, some authors used a subjective scale [8], peak concentric power output [15], or linear mean propulsive velocity [11] (corresponding to 90% of RM on ISO-load). Therefore, comparisons between loading condition and subsequent training effects might have led to misinterpretation of the results. An index that could help to relativize load intensity in FRTDs may be of help for this purpose. 

In 1994, Berg and Tesch [16] published the first research study employing an FRTD in RT. To date, the magnitude of the external FW-load (i.e., the flywheel moment of inertia) has been selected mainly based on arbitrary loads [12,13,17,18,19]. More recent works [20] based this comparison on an arbitrary, subjective perceived effort [21,22], or maximum peak power over a range of progressively increasing loads [23,24]. It is actually possible to normalize the load by using maximum peak power with reference to the optimal power zone concept [25]. At 100% of peak power output, the authors found that FRTD improved performance in change of direction [26], jump [26], sprint abilities [24,26], and throwing speed in handball [23]. In exercises executed under ISO-load, the loading training zones around the optimal power zone (i.e., peak power output load ± 20%) produced improvements in change of direction, jump, and sprint abilities in soccer players [27]. However, using a relative peak power value has an important limitation. Because the power–load or power-velocity relationship is typically parabolic [1], the same power value can be produced at different velocities or loads, thus resulting in different training effects. 

To determine the loading training intensity zone in ISO-load exercises, trainers typically use an incremental load test [2]. In recent years, some authors have focused on understanding the workload ranges in FRTD, especially in the squat exercise. Carroll et al. [28] found that an overload exists in a progressive loading test in FRTD, but with poor associations between the speed and moment of inertia in the concentric phase. In contrast, another work showed a better association between these variables, primarily when a logarithmic fit model was used individually (r2 = 0.97) [29]. In flywheel training, performance monitoring can be performed by measuring flywheel speed, from which other training variables can be derived. 

Recent research proposed angular acceleration as a sensitive parameter to differentiate between different moments of inertia in the leg extension exercise on an FRTD [30]. Therefore, the use of different mechanical variables can be explored to assess load intensity in FRTD. Consequently, the main aim of the current study was to find an index to characterize the loading intensity in a squat exercise performed on an FRTD. A secondary aim was to study the differences in the relative intensity workload spectrum between the FW-load and ISO-load. Based on previous works, our main hypothesis was that the relationship between angular acceleration and moment of inertia in an incremental load test might be of help in finding a maximum load index for FRTDs. 

## 2. Materials and Methods

### 2.1. Experimental Approach to the Problem

This study followed a randomized cross-over intervention design. Subjects executed two incremental load tests on the squat exercise, using flywheel (FW-load) or weights (ISO-load), separated by a one-week washout period. In total, subjects were involved in two familiarization sessions and two main intervention days. The familiarization sessions consisted of the execution of a squat exercise with both types of loading conditions, supervised by a strength and conditioning coach with previous experience in flywheel training.

### 2.2. Subjects

Twenty-five healthy and physically active males participated in the study (age: 22.9 ± 2.2 years, height: 1.8 ± 0.07 m, weight: 76.9 ± 8.2 kg). The subjects performed a minimum of 6 months of weekly resistance training involving lower body, and a minimum of 2 years of experience with squat exercises, but no prior experience using FRTD. The subjects were asked to avoid any kind of strenuous physical activity for a minimum of 48 h before each measurement, and not to change their daily nutritional or physical activity routines for the study. Considering the statistical test used (ANOVA within factors repeated measures design), a statistical power of 80%, an alpha error of 0.05, and an effect size of 0.43 (corresponding to a large ƞ^2^p), the minimum required sample size was 8 subjects. 

### 2.3. Procedures

Before any intervention, subjects performed a standardized warm-up consisting of 5 min cycling on a mechanical ergometer at a self-selected submaximal pace, followed by 5 min of lower limb mobilization, five countermovement jumps, and six submaximal repetitions on the squat device, using the lowest external load from the incremental load test (20 kg for ISO-load and 0.025 kg·m^2^ for FW-load).

#### 2.3.1. Incremental Load Test

The incremental load test consisted of the execution of five sets using a progressively increasing load during the squat exercise. The execution of the ISO-load test was similar to that for the FW-load (i.e., quick eccentric phase, followed by a short transition between eccentric and concentric phase, and, subsequently, a concentric phase executed at maximal voluntary effort). The subjects executed five repetitions per set and rested for five minutes between each set. The ISO-load test was executed on a Smith machine (Fitland, Spain), while the FW-load test was executed on an FRTD with a cylindrical flywheel shaft (kBox 3, Exxentric, AB Bromma, Sweden). During the ISO-load test, external loads were incremented based on the highest individual mean propulsive speed achieved during each set, as follows: load 1 ≈ 1.37 m/s-20% RM, load 2 ≈ 1.11 m/s-40% RM, load 3 ≈ 0.90 m/s-60% RM, load 4 ≈ 0.75 m/s-70% RM, and load 5 ≈ 0.64 m/s-80% RM. We used the relationship between mean propulsive speed and 1RM-% to stablish the relative loading and the 1RM for the ISO-load, based on the study of Sánchez-Medina et al., [2]. For the FW-load test, we increased the external loads with a starting inertia of 0.025 kg·m^2^, followed by increments of 0.025 kg·m^2^ until a maximum value of 0.125 kg·m^2^. For further analyses, the three repetitions with the highest mean concentric speed were selected in both tests. If any subject showed considerable variability between the mean concentric speed (i.e., denoting a bad execution technique), that load was not considered for further analyses. Only subjects that completed all of the five loads were considered in the final analyses (n = 21).

#### 2.3.2. Data Acquisition

We measured raw (instantaneous) speed and vertical force during exercise in both loading conditions. We measured vertical force from each leg with two single-axis force platforms (SmartCoach Europe AB, Stockholm, Sweden). During ISO-load sets, we measured linear vertical speed using a linear encoder (SmartCoach Power Encoder, SmartCoach Europe AB, Stockholm, Sweden). During FW-load exercises, we measured angular flywheel speed using a quadrature incremental rotary encoder (EMS22Q, Bourns, Riverside, CA, USA) connected to the flywheel shaft. All raw signals were captured simultaneously at a 100Hz sampling rate using a general-purpose, multichannel acquisition system prototype (SmartCoach MultiChannel, SmartCoach Technologies Inc., Pleasanton, CA, USA) and specific computer software (SmartPlot V4.7.0, SmartCoach Technologies Inc., Pleasanton, CA, USA).

We imported each raw data signal for each set into Matlab R2020a (The MathWorks Inc., Natic, MA, USA) for processing and analyses. First, we smoothed the raw force signal using a Butterworth second-order low-pass filter with a cut-off frequency of 1 Hz (the cut-off frequency was selected based on analyses of residuals). For the ISO-load, the multipurpose acquisition system smoothed the vertical speed using a 10-tap rolling average filter. A threshold speed of 0.05 m/s was used to detect each repetition, defining the concentric phase as the interval with positive speed. We smoothed the angular speed raw signal using a Butterworth second-order low-pass filter for the FW-load, with a cut-off frequency of 1 Hz (the cut-off frequency was selected based on analyses of residuals). We computed both linear (m/s^2^) and angular (rad/s^2^) accelerations, as the time derivative (incremental ratio) of filtered speed. Each repetition was detected during the FW-load condition using the change from negative (eccentric phase) to positive (concentric phase) acceleration values. For FW-load, we also calculated torque as the product of the flywheel moment of inertia times angular acceleration. Finally, we calculated power as the product of force (for ISO-load) or torque (for FW-load) and speed. We only used the concentric phase values for the analyses (propulsive phase (34) for ISO-load and whole concentric phase for FW-load).Before any intervention, subjects performed a standardized warm-up consisting of 5 min cycling on a mechanical ergometer at a self- selected submaximal pace, followed by 5 min of lower limb mobilization, five countermovement jumps, and six submaximal repetitions on the squat device, using the lowest external load from the incremental load test (20 kg for ISO-load and 0.025 kg·m^2^ for FW-load).

#### 2.3.3. Mechanical Variables

For the concentric phase, we calculated the mechanical variables commonly used in resistance training to control the training intensity, for each loading condition. Specifically for FW-load, we calculated the mean angular speed (MAS) and mean angular acceleration (MAA). Specifically for ISO-load, we calculated the mean linear speed (MLS) and the mean linear acceleration (MLA). Finally, for both loading conditions, we calculated the mean power (MP) and mean vertical force (MVF).

#### 2.3.4. Flywheel Workload Indexes

To find an index to characterize the loading intensity, we studied the associations between each mechanical variable and the moment of inertia used. We used four different regression models (see Statistical Analyses). Finally, in order to model the relationship between MAA and moment of inertia, we used a logarithmic model (y = a + b ln(x)). The logarithmic model showed one of the highest coefficients of determination (r2) (see results). Even if, as shown in physics, the acceleration on an FRTD asymptotically tends to zero for an infinite inertia, the goal was to find a theoretical finite inertia value corresponding to null acceleration. In this context, the logarithmic model ensures that such a value exists for any value of a and b.

This value (i.e., the intercept of the regression curve on the inertia (x-axis) is given by solving the logarithmic formula for y = 0, and represents the theoretical maximum inertia (Maximum Flywheel Load, MFL) corresponding to a null acceleration:(1)$$0=a+b\cdot \ln\left(\mathrm{MFL}\right)\mathrm{MFL}=e^{-a/b}$$

Consequently, it was also possible to compute the mathematical expression of mean torque (MT) as a function of the inertia (since torque = angular acceleration times moment of inertia), and, even more importantly, that allowed analytical computation of the value of inertia (load) at which torque is maximal. Omitting the intermediate calculation steps (see Appendix A), it was hence possible to obtain the analytical expression of the Peak Torque Load (PTL), i.e., the moment of inertia which corresponds to maximal torque:(2)$$\mathrm{PTL}=e^{-1}\cdot \mathrm{MFL}\;\approx 0.368\;\mathrm{MFL}$$

More details on the model, calculations, and mathematical demonstrations are provided in the Appendix A.

#### 2.3.5. Statistical Analyses

Data are shown as mean ± SD. We analyzed the individual associations between the moment of inertia and each mechanical variable, adjusting for four regression models: linear, quadratic polynomial, exponential, and logarithmic models. To study the relationships between both loading conditions, we analyzed the relationship between MFL and 1RM using the Pearson correlation coefficient (r) and its coefficient of determination (r2). Finally, to compare the mechanical differences between both loading conditions, we first normalized each mechanical variable to the individual peak variable obtained from the progressive squat test in each condition. Next, we normalized the data to compare the mechanical variables resulting from angular movement (FW-load) and linear movement (ISO-load). The mechanical differences were calculated using five arbitrary relative intensities (%MFL—FW-load or %1RM—ISO-load), thus representing a workload range. Mechanical differences were tested using a two-way repeated-measures test with two within-subject factors (workload range (5 factors) and workload loading condition (2 factors)). A simple main analyses was conducted using Intensity (i.e., workload rage) as a simple effect factor and Condition as a moderator factor if a significant interaction was observed. The magnitude of the interaction was quantified using the eta partial squared effect size (ƞ2p). An alpha level of 0.05 was used to determine the occurrence of a significant difference. All the analyses were conducted using JASP software for Windows (JASP Team, Version 0.14.1).

## 3. Results

### 3.1. Relationships between Mechanical Variables and Moment of Inertia

The associations between MAS-, MAA-, MP-, and MVF-moment of inertia are shown in Figure 1. The highest r2 was observed in MAS and MAA variables when the polynomial and logarithmic fit models were used. For MP and MVF variables, the polynomial fit model showed a higher r2.

### 3.2. Flywheel Training Intensity Index

MFL resulted in an average of 0.21 ± 0.05 kg·m^2^. Figure 2 shows the association between the MAA- and MT-%MFL. When MT was calculated using the logarithmic fit model between the MAA-moment of inertia association, the highest MT was observed at 36.8% ± 0.7 of MFL, in accordance with the theoretical model. Finally, the total PTL average resulted in an average of 0.08 ± 0.02 kg·m^2^. To illustrate the concept of this methodology, Figure 3 shows a comparison between subjects with a notable difference in both MFL and PTL. 

### 3.3. Relationships and Differences between Loading Conditions

There was a significant (*p* < 0.001) difference between 1RM and MFL (Figure 4). A significant interaction between Condition × Intensity was observed in all the mechanical variables (MAS ƞ2p = 0.259, *p* < 0.001; MAA ƞ2p = 0.237, *p* < 0.001; MP ƞ2p= 0.296, *p* < 0.001; MAS ƞ2p = 0.528, *p* < 0.001). The relative MAA (Figure 5B) and MP (Figure 5D) were higher in all the compared relative intensities for ISO-load. In contrast, the relative MAS (Figure 5A) showed higher relative values at 15, 30, 45, and 60%. In contrast, the relative MVF was higher at 45, 60, and 75% of the relative intensity for FW-load, while at 15% of relative intensity, it was higher for ISO-load (Figure 5C).

## 4. Discussion

This study introduced a novel index (MFL), to characterize and express in relative terms the load intensity in a squat exercise executed on an FRTD. In addition, the study compared the differences over the workload range of the squat exercise executed under flywheel and isoinertial conditions. Our results showed that MFL can be computed as the intercept of the logarithmic regression on MAA versus moment of inertia on FRTD training data. Hence, MFL represents the individual maximum possible load to be used. In addition, our results showed that the mechanical outcomes over the relative workload range differ in the squat exercise executed under ISO-load or FW-load conditions. 

MFL is a theoretical, abstract value, as it represents the ideal intercept of the acceleration–inertia curve on the x axis, i.e., the inertia at which the flywheel acceleration would be zero. From the physical law of FRTDs (Law of Newton for angular motion) it is known that (1):T = I·α → α = T/I (3)(3)
where α = angular acceleration (rad/s^2^), T = torque (N·m), and I = moment of inertia (kg·m^2^). 

Previous works showed differences between workload ranges in FRTD [29,30,31]. To our knowledge, only three previous studies have investigated the relationship between (linear) speed and moment of inertia, using linear [28,29,31] or logarithmic [29] regressions. Our results showed only loose individual fits for MP and MVF with any model. In contrast, MAS and MAA yielded r2 values corresponding to almost perfect fit (Figure 1), especially when second-degree polynomial, logarithmic, or exponential models were used. Our results agree with McErlain-Naylor and Beato (35), although they used linear instead of angular speed. In contrast, the association shown in our study is higher compared to that shown by Carroll et al. [28]. The main difference can be explained by the use of pooled data instead of individual data to calculate the regressions. Recent research showed that individual associations might result in higher accuracy than pooled data [32]. In addition, greater improvements were observed in power-oriented training [33] in ISO-load exercises. 

Although speed is currently the most commonly used variable to monitor loading intensity in real-time for ISO-load [2,10,11], power is more widely used in FRTD [4]. However, a recent work showed that angular acceleration is more sensitive to changes in load in leg extension exercise executed on an FRTD [30]. Therefore, the relationship between force and velocity (i.e., force-velocity profile) in the ISO-load squat [34] is used to determine the loading condition. In addition, from the force-velocity relationship, 1RM can be calculated as the x axis intercept [35]. Based on prior works and our results, MAA was chosen in our study to calculate an x axis intercept in order to define the MFL index (Figure 3). What is more, MAA showed a low inter-individual variation (Figure 1) and previous research suggested that MAA was a more sensitive candidate to distinguish among different loads [30]. 

Finding a possible index to express load in FRTDs in relative terms (i.e., as a function of a maximal load), already has immediate practical application per se. An interesting implication is the possibility of analytically computing the inertia (PTL) at which the exerted torque is maximized. The fact that its value is invariably equal to 36.8% MFL, ensures that it is always well in the range of applicability of the logarithmic model, where it reliably represents the physical behavior. Another practical implication of this novel methodology is that it allows the definition of different training load intensity zones based on either %MFL or %PTL. For the same purpose, in ISO-load the %1RM is widely used to define training zones focusing on maximal strength development (i.e., >85% 1RM) or muscle endurance (i.e., <30% 1RM) [1,36]. Likewise, MFL could be used to determine whether different relative loads result in different training effects. In agreement, Loturco et al. [27] showed that training within the optimum power zone resulted in a higher performance increment than a classic strength–power periodization. PTL could be used with a similar purpose in RT programs using FRTDs. Future studies could focus specifically on exploring these potential applications. 

The 1RM is also used in ISO-load exercises to discriminate subjects based on their force characteristics. For example, subjects with a 1RM higher than twice their body weight are considered strong individuals [37]. Our results showed a poor association between MFL and 1RM (Figure 4), suggesting that an individual with a good score in ISO-load exercises does not necessarily exhibit an equally good performance in FW-load. Our results also showed that the chosen workload range between both loading conditions does not match (Figure 5). When ISO-load is used, participants can produce higher speeds, accelerations, and power than in FW-load. In contrast, at medium loads (i.e., 45% MFL or RM), subjects can produce higher vertical forces. In particular, FRTD is known to produce high forces over the whole concentric phase [6]. For the first time, to our knowledge, we compared the differences in the relative loading continuum between both loading conditions, using MFL and RM. Consequently, the mechanical profile of the FW-load squat was completely different from the ISO-load squat. This might suggest that different training effects could be attained by using both technologies in squat exercise at similar relative load, using MFL and 1RM as indexes. Additionally, caution should be exercised when trying to match training loads in FW- and ISO-load, e.g., to compare their effectiveness in a training study. In ISO-load, higher relative load zones (i.e., >60% 1RM) induced greater hypertrophy compared to lower relative load zones (<30% 1RM), which conversely resulted in greater improvement in endurance [36]. Future studies using FRTDs should focus on understanding which relative load zones (based on either MFL or PTL) induce changes mainly in strength or endurance aspects. 

The proposed indexes have some limitations. First, the application of MFL or PTL to different exercises or types of FRTDs might yield different results. For example, Nuñez et al. [38] showed that the shape of the flywheel shaft (i.e., cylindrical or conical) determines different mechanical output at comparable moments of inertia. This is because for the same force, torque (=force times lever arm) can be vastly different with a different lever arm (i.e., the distance between the point of application of force and the flywheel axis, which in turn depends on the shaft diameter and shape). A second limitation of the MFL concept itself, is related to the maximum possible inertia attained on a fully loaded FRTD. On the device used in our study, the maximum inertia with the maximum number of installed flywheels is 0.200 kg·m^2^. Individuals with a higher MFL might not be able to work at a high relative load because it could exceed the machine’s load capability. This is not a problem per se, since the logarithmic model loses validity close to 100%MFL (see Appendix A for more information), and since the optimal load (PTL) is invariably achieved at ≈37% MFL. Therefore, it is still unknown if higher relative loads (e.g., >70–80% MFL) have practical interest and applications. Finally, the resolution in inertia adjustment is different for each FRTD, since the numbers and types of flywheel are usually limited. The device used in this study allowed for small increments down to 0.005 kg·m^2^. However, when the range of adjustment is limited, it might be necessary to adopt an available inertia different from the theoretically desired value. Future studies may analyze the reproducibility and validity of these indexes at different strength levels, in different exercises, and on different FRTD machines (e.g., with different shafts and loading ranges).

## 5. Conclusions

In conclusion, a logarithmic regression fitting model between MAA and moment of inertia allows calculation of both MFL and PTL indexes during exercise on an FRTD. Both indexes allow reliable expression of the training load intensity in relative terms, at least on the machine and (squat) exercise used in this study.

Using MFL to express a relative workload range compared to %1RM, the ISO-load squat exercise produced higher relative speed, acceleration, and power along the whole load spectrum. To calculate MFL in a FW-load squat exercise, an incremental load test can be executed using five equally spaced, different moments of inertia. In cases where no raw data or MAA are available, an approximate average acceleration can be computed by dividing the mean concentric speed by the duration of the concentric phase.

## Figures and Tables

**Figure 1 sensors-21-08124-f001:**
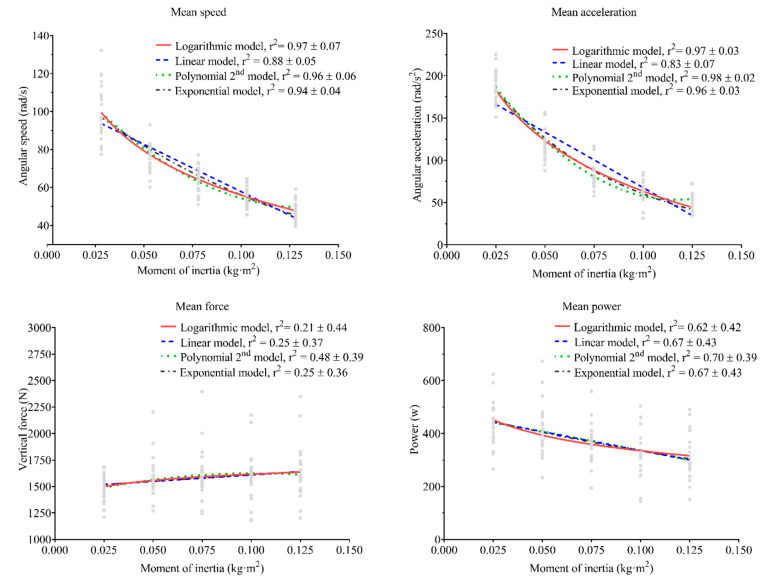
Association between each mechanical variable and the moment of inertia used, with different fitting models. r^2^ shows the mean ± standard deviation coefficient of determination for each model. The lines represent the regression line for each model, while the gray points represent individual values for each subject.

**Figure 2 sensors-21-08124-f002:**
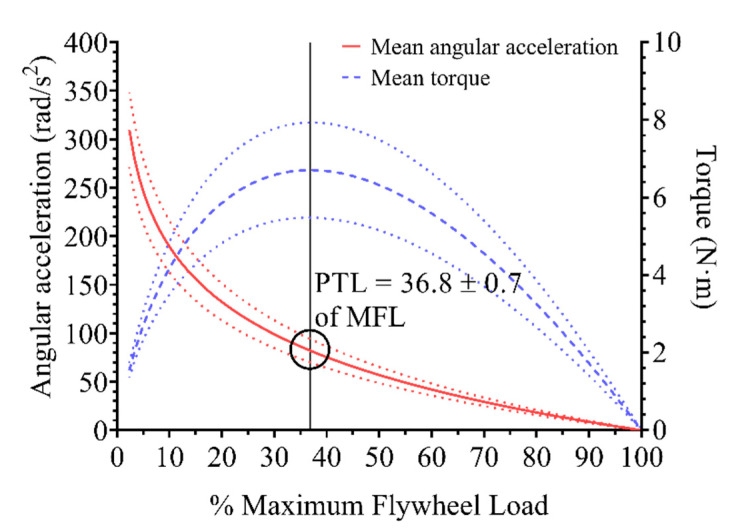
Association between mean angular acceleration and mean torque versus relative flywheel load. Lines represent mean ± standard deviation (dotted lines) of all the subjects’ individual regression lines (logarithmic fit for angular acceleration and second-degree polynomial for mean torque). The vertical black line represents the relative inertia at which peak torque load (PTL) is attained as a function of maximum flywheel load (MFL).

**Figure 3 sensors-21-08124-f003:**
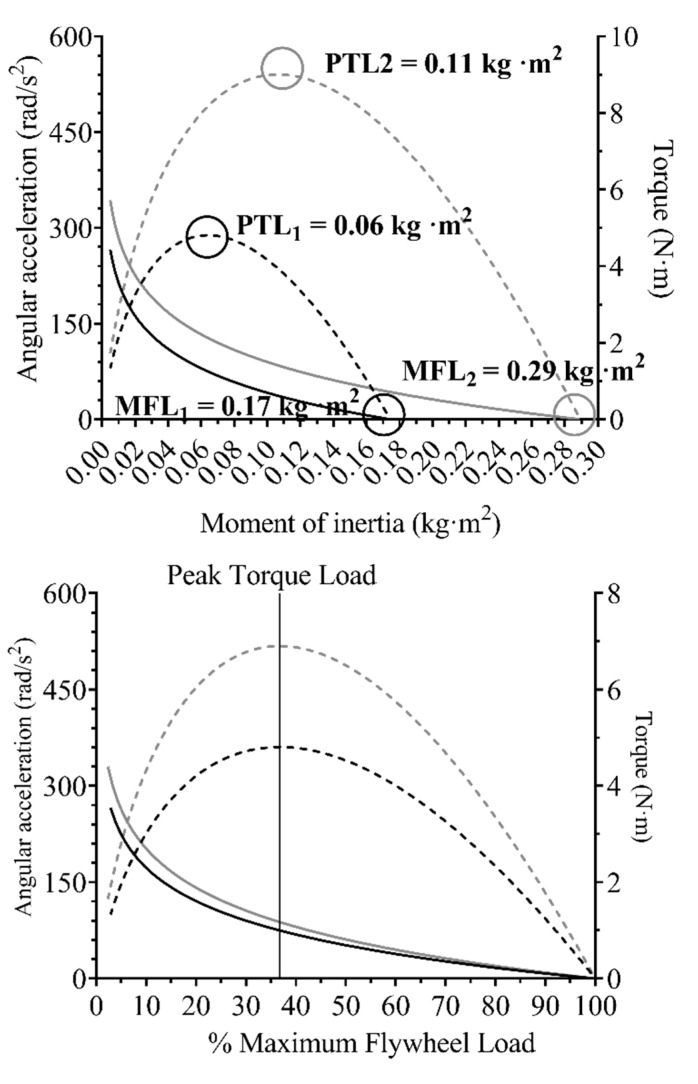
Illustration of the maximum flywheel load index concept. The upper graph shows the acceleration regression line (continuous line) and the torque regression line (dashed line) of a subject with a lower maximum flywheel load (MFL, black color) against a subject with a higher MFL (gray color). MFL is calculated as the intercept of the acceleration curve on the horizontal axis. The lower graph compares both subjects’ profiles using relative intensities instead.

**Figure 4 sensors-21-08124-f004:**
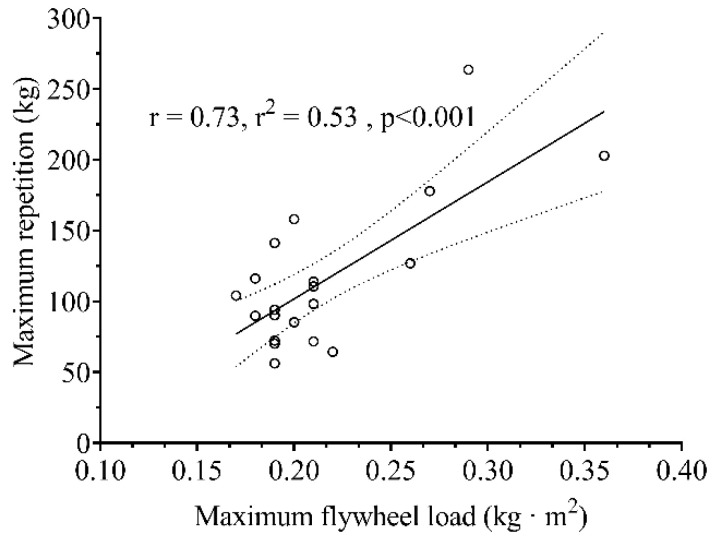
Association between maximum repetition (ISO-load) and the maximum flywheel load (FW-load). Dotted lines represent 95% of the confidence intervals.

**Figure 5 sensors-21-08124-f005:**
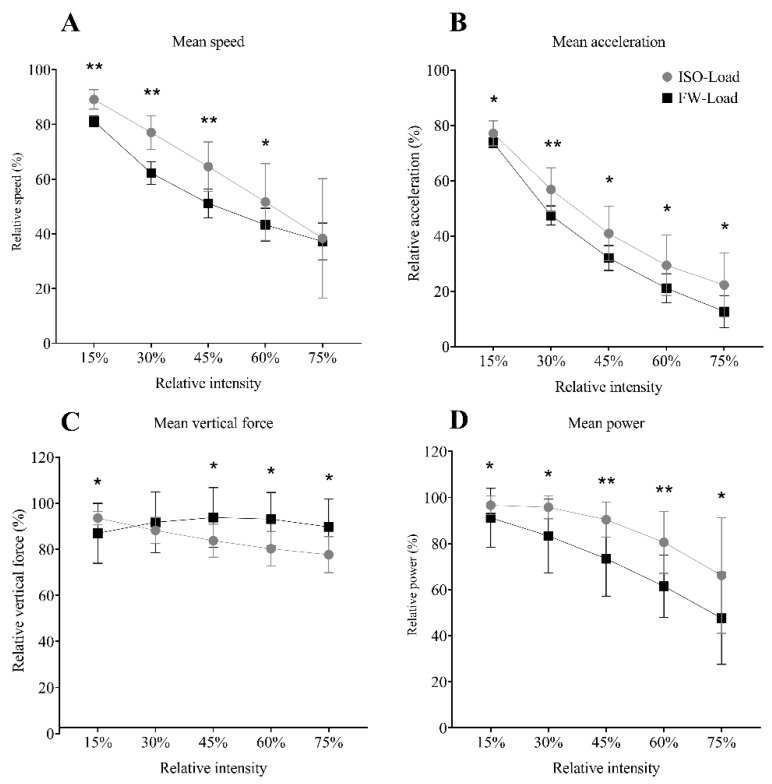
(**A**) Estimated data normalized from maximum speed; (**B**) acceleration; (**C**) vertical force, and (**D**) power for five relative intensities (training intensity continuum) for the flywheel (FW-load) and weight load types (ISO-load). Data points represent mean values, and error bars the standard deviation. Between loading condition differences are shown as: * = *p* < 0.05, ** = *p* < 0.001.

## Data Availability

Data available upon request.

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
