# Peer review of "The Maximum Flywheel Load: A Novel Index to Monitor Loading Intensity of Flywheel Devices"

_sensors, 2021, doi:10.3390/s21238124_

Round 1
Reviewer 1 Report
flywheel resistance training. The authors have provided a well written manuscript, based on a good idea, however I believe that the results provided, do not fit the aim of the study, which I suggest to modify accordingly. Further, some methodological considerations regarding the physical evaluation are warranted.
See below for specific comments.
Abstract
It doesn’t seem there is congruence between the title, the aim within the abstract and the introduction. Authors should clarify what the manuscript aims to.
Introduction
Why are the authors differentiating between RT and weight training?
The authors state “It is actually possible to normalize the load by using maximum peak power with reference to the optimal power zone concept”, however some lines after “However, currently, implementing relative load intensity training zones is unlikely in FRTD due to the absence of an index that could help”, can the authors please clarify?
Authors state that in FW training performance monitoring can be performed by measuring speed. Therefore, known the load, power can be easily calculated. Based on the above sentence and from the logarithmic fit of McErlain-Naylor et al, it seems that such approach would result easier for maximal load estimation in FW training.
The authors within the introduction do not talk of any specific exercises to assess 1RM or maximal FW-load, however in the introduction state “consequently, the main aim was” and introduce the squat. I would suggest either to introduce the rationale to the squat exercise or remove from the aim.
The authors state that the index based on angular acceleration and moment of inertia could help to find a maximum load index. The purpose of identifying indexes in this context is to allow practical and easy identification of loads in sport environments. It is highly improbable that in sporting environments people will be able to calculate neither moment of inertia or angular acceleration. Therefore, I am questioning if the detection of an index would really help in load identification.
Methods
It is not clear how come the authors did not perform a classical 1RM test to determine the iso-load. Why a 5x5 with incremental load was used? If the authors do not have a 1RM how can they determine thresholds of 20-40-60-70 and 80%RM?
Was speed for both assessment procedures pre-determined? In other words, did the authors instruct the participants regarding speed during the execution of the tests, or the participants were free to perform repetitions at their one pace?
In the statistical analysis section the authors state to have analyzed the relation between MFL and 1RM, however, the authors have not assessed the participants for 1RM.
Results
3.1 the authors should report the R2 values
As I understood, performance differences were present for all variables, at all intensities in either relative and absolute variables, between the two conditions. It is still not clear to me, based on the provided results, how the index can be applied in real life situations. In my opinion the authors should reconsider the aim the manuscript, by fitting it with the results provided.
Conclusions
If the MFL provided indexes which significantly differed from the %1RM, are the authors sure the participants were actually performing exercise in the FW at that specific intensity?
Thanks
Reviewer 2 Report
The paper clearly gives the environment, assumptions, requirements and objectives of the problem in hand, and points out major issues or difficulties when dealing with the problem and the model design. This work is interesting. However, it can be improved as follows:
1. This study introduced a novel index (MFL), to characterize and express the load intensity in a squat exercise executed on an FRTD. In addition, the study compared the differences over the workload range of the squat exercise executed under flywheel and iso-inertial conditions. Although the authors discussed the limitations of the proposed model, it would be better to provides some examples (e.g., compare and contrast the model with two different exercises) to further explore the characteristics of the proposed indexes.
2. Line 107: "height: 1.8 ± 0.7 m" or “height: 1.8 ± 0.07 m” ?
3. Since the index models are highly customized based on the subject features (e.g., age, height, weight, strength level, and physical condition) and exercise machines, please discuss sensible ways to expand the adjustment range of the index models.
Reviewer 3 Report
This manuscript describes a method to calculate a novel index that represents the work load of FRTD, and can be an equivalent to the one maximum repetition (1RM) weight in case of weight training. First, the authors experimentally found that the logarithmic regression can almost perfectly express the relationship between the mean angular acceleration and moment of inertia. Based on this finding, the authors then developed a calculation method to obtain a maximum flywheel load (MFL), which is a theoretical maximum inertia when angular acceleration is zero. Lastly they compared %MFL with %1RM, finding characteristic difference between the two indices. The proposed method is mathematically simple but seems to have meaningful influence on further application of FRTD. The reviewer recommends the publication of this manuscript in Sensors after minor revision based on the following comments:
- There are some editorial and grammatical errors in the figures and main body. Moreover, the resolutions of the figures are insufficient and the font sizes of the legends are too small.
- Why did not the authors show the equivalent to Figure 5 using %PTL. In case of %PTL, one %value represents two corresponding statuses of moment of inertia, because the MT line is not monotonically increasing nor decreasing against moment of inertial unlike MAA. Can this be a problem when PTL is used as a load intensity index?
Round 2
Reviewer 1 Report
The authors have addressed the majority of the concerns providing a revised version of the manuscript.
Despite the manuscript results improved, i still have some concern regarding some points.
Regarding RT and weight training i agree that the authors intended to be more clear, however i believe they did not manage to to so. I would suggest to return to the previous terminology (Resistance Training) and use so within the manuscript.
Author Response
Thank you for your time in reviewing again our paper.
We have changed again to Resistance Training at the beginning of the introduction. What is more, we have added a sentence to clarify that weight training (ISO-load in our case) is another option to develop resistance training. However, we think that we cannot use either one term or another because, as we explain, weight training is a part of resistance training options, such as flywheel another option for example.
The new sentence reads as When free weights (ISO-load) are used as an option to implement RT, exercise intensity can be measured efficiently by execution speed [2] (line 31).
Reviewer 2 Report
The authors have addressed some main concerns. To make this work complete, it would be better to provide some test examples (e.g., compare and contrast the model with two different exercises) to further explore the characteristics of the proposed indexes.
Author Response
Thank you for your time in reviewing again our paper. We agree with you and we have done a change accordingly, in line 393, as "For example, a seated rowing exercise executed on a conical pulley and a cylindrical flywheel device [39], executed on individuals with different levels of 1RM, could highlight the usefulness of the MFL index for the proposed purposes"